# Neural AMR:
# Sequence-to-Sequence Models for Parsing and Generation

## Abstract

While sequence-to-sequence (seq2seq) models have been broadly used, their application to Abstract Meaning Representation (AMR) parsing and AMR realization has been limited, at least in part because data sparsity was thought to pose a significant challenge. In contrast, we show that with careful preprocessing and a novel training procedure that allows us to incorporate millions of unlabeled sentences, we can significantly reduce the impact of sparsity. For parsing, we obtain competitive results of 61.9 SMATCH, the current best reported without significant use of external annotated semantic resources. For realization, we outperform state of the art by over 5 points, achieving 32.3 BLEU. We also present extensive ablative and qualitative analysis in addition to showing strong evidence that seq2seq models are robust to artifacts introduced by converting AMR graphs to sequences.

## 1   Introduction

Abstract Meaning Representation (AMR) is a graph based formalism that encodes many aspects of the meaning of a natural language sentence (for example, see Figure 1). AMR has been used as an intermediate representation for machine translation (Jones et al., 2012), summarization (Liu et al., 2015), sentence compression (Takase et al., 2016), event extraction (Huang et al., 2016), and has potential applications in dialogue, or human-robotic interaction. While AMR is extremely expressive, annotation is expensive and training data is limited, making application of neural methods challenging (Misra and Artzi, 2016; Peng and Xue, 2017; Barzdins and Gosko, 2016).

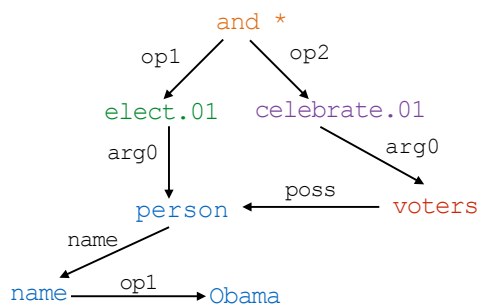

Figure 1: An example sentence and its corresponding Abstract Meaning Representation (AMR). AMR encodes semantic dependencies between entities mentioned in the sentence, such as "Obama" being the "arg0" of the verb "elected".

In this work, we tackle both AMR parsing and AMR realization together, showing the first successful sequence-to-sequence (seq2seq) models for both problems. While seq2seq models have been broadly used (Wu et al., 2016; Bahdanau et al., 2014; Luong et al., 2015; Vinyals et al., 2015), application to AMR has been limited, at least in part because effective *linearization* (encoding graphs as linear sequences) and data sparsity were thought to pose significant challenges. We show these challenges can be easily overcome, by demonstrating that seq2seq models can be trained using **any** graph-isomorphic linearization and that unlabeled text can be used to significantly reduce sparsity.

Our approach is two-fold. First, we carefully preprocess the AMR, anonymizing entities and dates, grouping entity categories, and encoding nesting information in concise ways, for example see Figure 2(d). Under such a representation, we show that any depth first traversal of the AMR is effective for constructing a linearization, and it is even possible to use a different random or-

der for each example. Second, we introduce a novel paired training procedure (Algorithm 1) for constructing an AMR parser and realizer. First we use self-training to bootstrap a high quality AMR parser from millions of unlabeled Gigaword (Napoles et al., 2012) sentences and then use it to pre-train an AMR realizer. The paired training allows both the parser and realizer to learn high quality representations of input and output language, respectively, from millions of weakly labeled examples, that are then further improved by training on human annotated AMR data.

Experiments on the LDC2015E86 AMR corpus (SemEval-2016 Task 8) demonstrate the effectiveness of the overall approach. For parsing, we are able to obtain competitive performance of 61.9 SMATCH without using any external annotated examples other than the output of a NER system, an improvement of over 9 points relative to neural models with a comparable setup. For realization, we are able to substantially outperform state of the art, providing gains of over 5 points, for a final performance of 32.30 BLEU. We also provide extensive ablative and qualitative analysis, quantifying the contributions that come from preprocessing and the paired training procedure.

## 2 Related Work

**Alignment-based Parsing** Flanigan et al. (2014) (JAMR) pipeline concept and relation identification with a graph-based algorithm that attempts to find the maximum spanning sub-graph. Zhou et al. (2016), extend JAMR by performing the concept and relation identification tasks jointly with an incremental model. Both systems rely on features based on a set of alignments produced using bi-lexical cues and hand-written rules. In contrast, our models train directly on parallel corpora, and make only minimal use of alignments to anonymize named entities.

**Grammar-based Parsing** Wang et al. (2016) (CAMR) perform a series of shift-reduce transformations on the output of an externally-trained dependency parser, similar to Brandt et al. (2016),Puzikov et al. (2016), and Goodman et al. (2016). Artzi et al. (2015) use a grammar induction approach with Combinatory Categorical Grammar (CCG), which relies on pre-trained CCGBank categories, like Bjerva et al. (2016). Pust et al. (2015) recast parsing as a string-to-tree Machine Translation problem, using unsupervised alignments (Pourdamghani et al., 2014), and employing several external semantic resources. Our neural approach is engineering lean, relying only on a large unannotated corpus of English and algorithms to find and canonicalize named entities.

**Neural Parsing** Recently there have been a few seq2seq systems for AMR parsing (Barzdins and Gosko, 2016; Peng and Xue, 2017). Similar to our approach, Peng and Xue (2017) deal with sparsity by anonymizing named entities and typing low frequency words, resulting in a very compact vocabulary (2k tokens). We avoid reducing our vocabulary by introducing a large set of unlabeled sentences from an external corpus.

**AMR Realization** Flanigan et al. (2016) specify a number of tree-to-string transduction rules based on alignments and POS-based features, that are used to drive a tree-based SMT system. Pourdamghani et al. (2016) also use an MT decoder; they learn a classifier that linearizes the input AMR graph in an order that follows the output sentence, effectively reducing the number of alignment crossings of the phrase-based decoder. Song et al. (2016) recast generation as a traveling salesman problem, after partitioning the graph into fragments, and finding the best linearization order. Our models do not need to rely on a particular linearization of the input, attaining comparable performance even with a per example random traversal of the graph. Finally, all three systems intersect with a large LM trained on Gigaword. We show that our seq2seq model has the capacity to learn the same information as a language model, especially after pretraining on the external corpus.

**Data Augmentation** Our paired training procedure is largely inspired by Sennrich et al. (2016). They improve neural MT performance for low resource language pairs, by creating synthetic output for a large monolingual corpus of the target language, using a back-translation MT system and mix it with the human translations. We instead pre-train on the external corpus first, and then fine-tune on the original dataset.

## 3 Methods

In this section we describe: (1) the tasks of AMR parsing and realization formally, (2) the form of sequence-to-sequence models we use, and (3) our paired training procedure, Algorithm 1.

## 3.1 Tasks

We assume access to training dataset $D$ where each example pairs a natural language sentence $s$ with an AMR $a$. The AMR is a rooted directed acylical graph. It contains nodes whose names correspond to sense-identified verbs, nouns, or AMR specific concepts, for example `elect.01`, `Obama`, and `person` in Figure 1. One of these nodes is a distinguished root, for example, the node `and` in Figure 1. Furthermore, the graph contains labeled edges, which correspond to ProbBank-style (Palmer et al., 2005) semantic roles for verbs or other relations introduced for AMR, for example, `arg0` or `op1` in Figure 1. The set of node and edge names in an AMR graph is drawn from a set of concepts $C$, and every token in a sentence is drawn from a vocabulary $W$.

We study the task of training an **AMR parser**, i.e., finding a set of parameters $\theta_P$ for model $f$, that predicts an AMR graph $\hat{a}$, given a sentence $s$:

$$\hat{a} = \underset{a}{\operatorname{argmax}} f\big(a|s;\theta_P\big) \qquad (1)$$

We also consider the reverse task, training an **AMR realizer** by finding a set of parameters $\theta_G$, for a model $f$ that predicts a sentence $\hat{s}$, given an AMR graph $a$:

$$\hat{s} = \underset{s}{\operatorname{argmax}} f\big(s|a;\theta_G\big) \qquad (2)$$

In both cases, we use the same family of predictors $f$, sequence-to-sequence models that use global attention, but the models have independent parameters , $\theta_P$ and $\theta_G$.

## 3.2 Sequence-to-sequence Model

For both tasks, we use a stacked-LSTM sequence-to-sequence neural architecture employed in neural machine translation (Bahdanau et al., 2014; Wu et al., 2016). Our model uses a global attention decoder and unknown word replacement with small modifications (Luong et al., 2015).

The model uses a stacked bidirectional-LSTM encoder to encode an input sequence and a stacked LSTM to decode from the hidden states produced by the encoder. We make two modifications to the encoder: (1) we concatenate the forward and backward hidden states at every level of the stack instead of at the top of the stack, and (2) introduce dropout in the first layer of the encoder. The decoder predicts an attention vector over the encoder hidden states using previous decoder states.

---

**Algorithm 1** Paired Training Procedure

---

**Input:** Training set of sentences and AMR graphs $(s, a) \in \mathcal{D}$, an unannotated external corpus of sentences $S_e$, a number of self training iterations, $N$, and a initial sample size $k$.
**Output:** Model parameters for AMR parser $\theta_P$ and AMR realizer $\theta_G$.
 1: $\theta_P \leftarrow$ Train parser on D
 ▷ Self-train AMR parser.
 2: $S_e^1 \leftarrow$ sample $k$ sentences from $S_e$
 3: **for** $i = 1$ to $N$ **do**
 4: $A_e^i \leftarrow$ Parse $S_e^i$ using parameters $\theta_P$
 ▷ Pre-train AMR parser.
 5: $\theta_P \leftarrow$ Train parser on $(A_e^i, S_e^i)$
 ▷ Fine tune AMR parser.
 6: $\theta_P \leftarrow$ Train parser on D with initial parameters $\theta_P$
 7: $S_e^{i+1} \leftarrow$ sample $k \cdot 10^i$ new sentences from $S_e$
 8: **end for**
 9: $S_e^N \leftarrow$ sample $k \cdot 10^N$ new sentences from $S_e$
 ▷ Pre-train AMR realizer.
10: $A_e \leftarrow$ Parse $S_e^N$ using parameters $\theta_P$
11: $\theta_G \leftarrow$ Train realizer on $(A_e^N, S_e^N)$
 ▷ Fine tune AMR realizer.
12: $\theta_G \leftarrow$ Train realizer on D using initial parameters $\theta_G$
13: **return** $\theta_P, \theta_G$

---

The attention is then used to weight the hidden states of the encoder and then predict a token in the output sequence. The weighted hidden states, the decoded token, and an attention signal from the previous time step (input feeding) are then fed together as input to the next decoder state. The decoder can optionally choose to output an unknown word symbol, in which case the predicted attention is used to copy a token directly from the input sequence into the output sequence.

## 3.3 Linearization

Our seq2seq models require that both the input and target be presented as a linear sequence of tokens. We define a linearization order for an AMR graph as any sequence of its nodes and edges (potentially with repeats). A linearization is defined as (1) a linearization order and (2) a rendering function that generates any number of tokens when applied to an element in the linearization order. Furthermore, for parsing, a valid AMR graph must be recoverable from the linearization.

## 3.4 Paired Training

Obtaining a corpus of jointly annotated pairs of sentences and AMR graphs is expensive and current datasets only extend to thousands of examples. Neural sequence-to-sequence models suffer from sparsity with so few training pairs. To reduce the effect of sparsity, we use an external unannotated corpus of sentences $S_e$, and a procedure which pairs the training of the parser and realizer.

Our procedure is described in Algorithm 1, and first trains a parser on the dataset $D$ of pairs of sentences and AMR graphs. Then it uses self-training to improve the initial parser. Every iteration of self-training has three phases: (1) parsing samples from a large, unlabeled corpus $S_e$, (2) creating a new set of parameters by training on $S_e$, and (3) fine-tuning those parameters on the original paired data. After each iteration, we increase the size of the sample from $S_e$ by an order of magnitude. After we have the best parser from self-training, we use it to label AMRs for $S_e$ and pre-train the realizer. The final step of the procedure fine-tunes the realizer on the original dataset $D$.

## 4 Data Handling

In the following sections, we explain a series of data preparation steps, including AMR linerization, anonymization, and other modifications we make to sentence-graph pairs to produce input and output for our seq2seq model. Our methods have two goals: (1) reduce the length of sentences and linearizations to make learning easier while maintaining enough original information, and (2) address sparsity from certain open class vocabulary entries, such as named entities (NEs) and quantities. Figure 2(d) contains example inputs and outputs with all of our preprocessing techniques.

**Basic Preprocessing** In order to reduce the overall length of the linearized graph, we first remove variable names and the `instance-of` relation ( `/` ) before every concept. In case of re-entrant nodes we replace the variable mention with its co-referring concept. Even though this replacement incurs loss of information, often the surrounding context helps recover the correct realization, e.g., the possessive role `:poss` in the example of Figure 1 is strongly correlated with the surface form *his*. Following Pourdamghani et al. (2016) we also remove senses from all concepts for AMR realization only. Figure 2(a) contains an example output after this stage.

### 4.1 Anonymization of Named Entities

Open-class types including NEs, dates, and numbers account for the 9.6% of tokens in the sentences of the training corpus, and 31.22% of the NL vocabulary. 83.36% of them occur fewer than 5 times in the dataset. In order to reduce sparsity and be able to account for new unseen entities, we perform extensive anonymization.

First, we anonymize sub-graphs headed by one of AMR's over 140 fine-grained entity types that contain a `:name` role. This captures structures referring to entities such as `person`, `country`, miscellaneous entities marked with `*-enitity`, and typed numerical values, `*-quantity`. We exclude `date` entities (see the next section). We then replace these sub-graphs with a token indicating fine-grained type and an index, $i$, indicating it is the $i$th occurrence of that type [1]. For example, in Figure 2 the sub-graph headed by `country` gets replaced with `country_0`.

On the training set, we use JAMR and unsupervised alignments to find mappings of anonymized subgraphs to spans of text (Flanigan et al., 2014; Pourdamghani et al., 2014) and replace mapped text with the anonymized token that we inserted into the AMR graph. We record this mapping for use during testing of realization models. If a realization model predicts an anonymization token, we find the corresponding token in the AMR graph and replace the model's output with the most frequent mapping observed during training for the entity name. If the entity was never observed, we copy its name directly from the AMR graph.

**Anonymizing Dates** For dates in AMR graphs, we use separate anoymization tokens for year, month-number, month-name, day-number and day-name, indicating whether the date is mentioned by word or by number.[2] In AMR realization, we render the corresponding format when predicted. Figure 2(b) contains an example of all preprocessing up to this stage.

**Named Entity Clusters** When performing AMR realization, each of the AMR fine-grained entity types is manually mapped to one of the four coarse entity types. We adopt the same types as used in the Stanford NER system (Finkel et al., 2005): person, location, organization and misc. This reduces the sparsity associated with many rarely occurring entity types. Figure 2 (c) contains an example with named entity clusters.

**NER for Parsing** When parsing, we must normalize test sentences to match our anonymized training data. To produce fine-grained named entities, we run the Stanford NER system (Finkel

---

[1] In practice we only used three groups of ids: a different one for NEs, dates and constants/numbers.

[2] We additionally use three date format markers that appear in the text as: *YYYYMMDD*, *YYMMDD*, and *YYYY-MM-DD*.

US officials held an expert group meeting in **January 2002** in **New York**.

```
(h / hold-04
    :ARG0 (p2 / person
        :ARG0-of (h2 / have-org-role-91
            :ARG1 (c2 / country
                :name (n3 / name
                    :op1 "United" op2: "States"))
            :ARG2 (o / official)))
    :ARG1 (m / meet-03
        :ARG0 (p / person
            :ARG1-of (e / expert-01)
                :ARG2-of (g / group-01)))
    :time (d2 / date-entity :year 2002 :month 1)
    :location (c / city
        :name (n / name :op1 "New" :op2 "York")))
```

**(a)** US officials held an expert group meeting in **January 2002** in **New York**.
```
hold
    :ARG0 person :ARG0-of have-org-role :ARG1 country :name name :op1
United :op2 States :ARG2 official
    :ARG1 meet :ARG0 person :ARG1-of expert :ARG2-of group
    :time date-entity :year 2002 :month 1
    :location city :name name :op1 New :op2 York
```

**(b)** **country_0** officials held an expert group meeting in **month_0 year_0** in **city_1**.
```
hold
    :ARG0 person :ARG0-of have-org-role :ARG1 country_0 :ARG2 official
    :ARG1 meet :ARG0 person :ARG1-of expert :ARG2-of group
    :time date-entity year_0 month_0
    :location city_1
```

**(c)** **loc_0** officials held an expert group meeting in **month_0 year_0** in **loc_1**.
```
hold
    :ARG0 person :ARG0-of have-org-role :ARG1 loc_0 :ARG2 official
    :ARG1 meet :ARG0 person :ARG1-of expert :ARG2-of group
    :time date-entity year_0 month_0
    :location loc_1
```

**(d)** **loc_0** officials held an expert group meeting in **month_0 year_0** in **loc_1**.
```
hold
    :ARG0 ( person :ARG0-of ( have-org-role :ARG1 loc_0 :ARG2 official ) )
    :ARG1 ( meet :ARG0 ( person :ARG1-of expert :ARG2-of group ) )
    :time ( date-entity year_0 month_0 )
    :location loc_1
```

Figure 2: Preprocessing methods applied to sentence (top row) - AMR graph (left column) pairs. Sentence-graph pairs after (a) basic preprocessing, (b) named entity anonymization, (c) named entity clustering, and (d) insertion of scope markers.

et al., 2005) and first try to replace any identified span with a fine-grained category based on alignments observed during training. If this fails, we anonymize the sentence using the coarse categories predicted by the NER system, which are also categories in AMR. After parsing, we deterministically generate AMR for anonymizations using the corresponding text span.

### 4.2 Linearization

A linearization is specified by (1) a linearization order, a list of nodes and edges to visit, possibly with repeats and (2) a rendering function that emits any number of tokens given an element in the linearization order.

**Linearization Order** Our linearization order is analogous to the order of nodes visited by depth first search, including backward traversing steps. For example, in Figure 2, starting at `meet` the order contains `meet`, `:ARG0`, `person`, `:ARG1-of`, `expert`, `:ARG2-of`, `group`, `:ARG2-of`, `:ARG1-of`, `:ARG0`.[3] The order traverses children in the sequence they are presented in the AMR. We consider alternative orderings of children in Section 7 but always follow the pattern demonstrated above.

**Rendering Function** Our rendering function marks scope, and generates tokens based on two cases. (1) If the element is a node, it emits the type of the node. (2) If the element is an edge, the first time an edge is encountered, if the edge leads to a node with children it emits a left parenthesis " (" and then emits the type of the edge. The second time an edge is encountered, if a " (" was emitted previously, a right parenthesis " )" is emitted. This rendering function omits our scope markers " (" and " )" in cases when a node only has one child, significantly reducing the number of tokens it generates. Figure 2(d) contains an example showing all of the preprocessing techniques and scope markers that we use in our full model.

## 5 Experimental Setup

We conduct all experiments on the AMR corpus used in SemEval-2016 Task 8 (LDC2015E86), which contains 16,833/1,368/1,371 train/dev/test examples. For the paired training procedure of Algorithm 1 we use Gigaword as our external corpus and sample sentences that only contain tokens from the AMR corpus. We evaluate AMR parsing with SMATCH (Cai and Knight, 2013), and AMR realization using BLEU (Papineni et al., 2002) [4].

We validated word embedding sizes and RNN hidden representation sizes by maximizing development performance just using data in the AMR corpus, Algorithm 1, line 1. We searched over the set {128, 256, 500, 1024} for the best combinations of sizes and set both to 500. Models were trained by optimizing cross-entropy loss with

---

[3] Sense and `instance-of` information has been removed at the point of linearization

[4] We use the multi BLEU script.

| Model | Prec | Rec | F1 | Prec | Rec | F1 |
|---|---|---|---|---|---|---|
| SBMT (Pust et al., 2015) | - | - | 69.0 | - | - | 67.1 |
| CAMR (Wang et al., 2016) | 72.3 | 61.4 | 66.6 | 70.4 | 63.1 | 66.5 |
| CCG* (Artzi et al., 2015) | 67.2 | 65.1 | 66.1 | 66.8 | 65.7 | 66.3 |
| JAMR (Flanigan et al., 2014) | - | - | - | 64.0 | 53.0 | 58.0 |
| GIGA-2M | 61.9 | 64.8 | **63.3** | 60.2 | 63.6 | **61.9** |
| GIGA-200k | 59.7 | 62.9 | 61.3 | 57.8 | 60.9 | 59.3 |
| AMR-ONLY | 54.9 | 60.0 | 57.4 | 53.1 | 58.1 | 55.5 |
| SEQ2SEQ (Peng and Xue, 2017) | - | - | - | 55.0 | 50.0 | 52.0 |
| CHAR-LSTM (Barzdins and Gosko, 2016) | - | - | - | - | - | 43.0 |

Table 1: SMATCH scores for AMR Parsing (TEST set). *Reported numbers are on the newswire portion of a previous release of the corpus (LDC2014T12).

| Model | Dev | Test |
|---|---|---|
| GIGA-2M | **31.8** | **32.3** |
| GIGA-200k | 27.2 | 27.4 |
| AMR-ONLY | 21.7 | 22.0 |
| PBMT* (Pourdamghani et al., 2016) | 27.2 | 26.9 |
| TSP (Song et al., 2016) | 21.1 | 22.4 |
| TREETOSTR (Flanigan et al., 2016) | 23.0 | 23.0 |

Table 2: BLEU results for AMR Realization. *Model has been trained on a previous release of the corpus (LDC2014T12).

| Model | BLEU |
|---|---|
| Full | 21.76 |
| Full - scope | 19.68 |
| Full - scope - ne | 19.53 |
| Full - scope - ne - anon | 18.71 |

Table 3: BLEU scores for AMR realization ablation results on preprocessing (DEV set).

stochastic gradient descent, using a batch size of 100, and dropout rate of 0.5. Across all models when performance does not improve on the AMR dev set, we decay the learning rate by 0.8.

For the initial parser trained on the AMR corpus, Algorithm 1, line 1, we use a single stack version of our model, set initial learning rate to 0.5 and train for 60 epochs, taking the best performing model on the development set. All subsequent models benefited from increased depth and we used 2-layer stacked versions, maintaining same embedding sizes. We set the initial Gigaword sample size to $k = 200,000$ and executed a maximum of 2 iterations of self-training. For pre-training the parser and realizer, Algorithm 1, lines 4 and 9, we used an initial learning rate of 1.0, and ran for 20 epochs. We attempt to fine-tune the parser and realizer, respectively, after every epoch of pre-training, setting the initial learning rate to 0.1. We select the best performing model on the development set among all of these fine-tuning attempts.

## 6  Results

**Parsing Results**  Table 1 summarizes our development results for different rounds of self-training and test results for our final system, self-trained on 2M unlabeled Gigaword sentences. Through every round of self-training, our parser improves. Our final parser outperforms comparable seq2seq and character LSTM models by over 9.8 points. While much of this improvement comes from self-training, our model without Gigaword data outperforms these approaches by 5.4 points on F1. All other models that we compare against use semantic resources, such as WordNet, dependency parsers or CCG parsers (models marked with * were trained with less data, but only evaluate on newswire text; the rest evaluate on the full test set, containing text from blogs). Our full models outperform JAMR, a graph-based model but still lags behind other parser-dependent systems (CAMR), and resource heavy approaches (SBMT).

**Realization Results**  Table 2 summarizes our AMR realization results on development and test set. We outperform all previous state-of-the-art systems by the first round of self-training and further improve with the second round. Our final model trained on GIGA-2M outperforms previous models by 5.4 BLEU. Overall, our model incorporates less data than previous approaches as all reported methods train language models on the whole Gigaword corpus. We leave scaling our models to all of Gigaword for future work.

| Model | Prec | Rec | F1 |
|---|---|---|---|
| Full | 54.9 | 60.0 | 57.4 |
| Full - anon | 22.7 | 54.2 | 32.0 |

Table 4: SMATCH scores for AMR parsing ablation results on preprocessing (DEV set).

| Linearization Order | BLEU |
|---|---|
| Human | 21.67 |
| Random | 20.80 |
| Stochastic | 20.34 |

Table 5: BLEU on AMR realization for different linearization orders (DEV set).

**Preprocessing Ablation Study** We consider the contribution of each main component of our pre-processing stages while keeping our linearization order identical. Figure 2 contains examples of linearized AMR and sentence pairs we evaluate for each setting of our ablations. First, we evaluate with AMR linearized without parentheses for indicating scope, Figure 2(c), then additionally without named entity clusters, Figure 2(b), and additionally without any anonymization, Figure 2(a).

Tables 3 summarizes our evaluation on the AMR realization. Results indicate each of these components is required, and that scope markers and anonymization are the biggest contributors. We suspect without scope markers our seq2seq models are not as effective at capturing long range semantic relationships between elements of the AMR graph. We also evaluated the contribution of anonymization to AMR parsing, Table 4.Similar to previous work, we find seq2seq based AMR parsing is largely ineffective without anonymization (Peng and Xue, 2017).

# 7 Linearization

In this section we evaluate three strategies for converting AMR graphs into sequences in the context of AMR realization and show that our models are largely agnostic to linearization orders. Our results argue, unlike SMT-based AMR realization methods (Pourdamghani et al., 2016), that seq2seq models can learn to ignore artifacts of the conversion of graphs to linear sequences.

## 7.1 Linearization Orders

All linearizations we consider use the pattern described in Section 4.2, but differ on the order in which children are visited. Each linearization generates anonymized, scope marked output (see Section 4), of the form in Figure 2(d).

**Human** The proposal traverses children in the order presented by human authored AMR annotations, exactly as shown in Figure 2(d).

**Random** We construct a random global ordering of all edge types appearing in AMR graphs. We traverse children, based on the position in the global ordering of the edge leading to a child.

**Stochastic** In this linearization we randomize our traversal of children, per example.

## 7.2 Results

We present AMR realization results for the three proposed realization orders in Table 5. Random linearization order performs only slightly worse than traversing the graph according to Human linearization order. Surprisingly, a per example stochastic linearization order performs nearly identically to a stable random order, arguing seq2seq models can learn to ignore artifacts of the conversion of graphs to linear sequences.

**Human-authored AMR leaks information** The small difference between stochastic and random linearizations argues that our models are largely agnostic to variation in linearization order. On the other hand, the model that follows the human order performs significantly better which leads us to suspect it carries extra information not apparent in the graphical structure of the AMR.

To further investigate, we compared the relative ordering of edge pairs under the same parent to relative position of children nodes derived from those edges, in a sentence, as reported by JAMR alignments. We found that the majority of pairs of AMR edges, 57.6%, always occurred in the same relative order, therefore revealing no extra realization order information.[5] Of the examples corresponding to edge pairs that showed variation, 70.3% appeared in an order consistent with the order they were realized in the sentence. The relative ordering of some pairs of AMR edges was particularly indicative of realization order. For example, the relative ordering of edges with types `location` and `time`, was 17% more indicative of the realization order than the majority of realizing locations before `time`.[6]

---

[5]This is consistent with constraints encoded in the annotation tools used to collect AMR. For example, `arg0` edges are always ordered before `arg1` edges

[6]Consider the sentences *"She went to school in New York*

| Error Type | % |
|---|---|
| Coverage | 29 |
| Disfluency | 23 |
| Anonymization | 14 |
| Sparsity | 13 |
| Attachment | 12 |
| Other | 10 |

Table 6: Error analysis for AMR realization on a sample of 50 examples from the development set.

To compare to previous work we still report using human orderings. However, we note that any practical application requiring a system to generate an AMR representation with the intention to realize it later on, e.g., a dialogue agent, will need to be trained either using consistent, or stochastic-derived linearization orders. Arguably, our models are agnostic to this choice.

## 8 Qualitative Results

Figure 3 shows example output of our realizer.The realization of the first graph is nearly perfect with only a small grammatical error due to anonymization. The second example is more challenging, with a deep right-branching structure, and a coordination of the verbs `stabilize` and `push` in the subordinate clause headed by `state`. The model omits some information from the graph, namely the concepts `terrorist` and `virus`. In the third example there are greater parts of the graph that are missing, such as the whole subgraph headed by `expert`. Also the model makes wrong attachment decisions in the last two subgraphs (it is the `evidence` that is *unimpeachable* and *irrefutable*, and not the `equipment`), mostly due to insufficient annotation (`thing`) thus making their realization harder.

Finally, Table 6 summarizes the proportions of error types we identified on 50 randomly selected examples from the development set. We found that the realizer mostly suffers from coverage issues, an inability to mention all tokens in the input, followed by fluency mistakes, as illustrated above. Attachment errors are less frequent, which supports our claim that the model is robust to graph linearization, and can successfully encode long range dependency information between concepts.

*two years ago"*, and *"Two years ago, she went to school in New York"*, where *"two year ago"* is the time modifying constituent for the verb *went* and *"New York"* is the location modifying constituent of *went*.

```
limit
 :arg0 ( treaty :arg0-of ( control :arg1 arms ) )
 :arg1 ( number
 :arg1 ( weapon :mod conventional
  :arg1-of ( deploy
   :arg2 ( relative-pos :op1 loc_0 :dir west )
  :arg1-of possible ) ) ) )
```
REF: the arms control treaty limits the number of conventional weapons that can be deployed west of the Ural Mountains .

SYS: the arms control treaty limits the number of conventional weapons that can be deployed west **of Ural Mountains** .

COMMENT: **disfluency**

```
state
 :arg0 report
 :arg1 ( obligate :arg1 ( government-organization
  :arg0-of ( govern :arg1 loc_0 ) )
  :arg2 ( help :arg1 ( and
  :op1 ( stabilize :arg1 ( state :mod weak ) )
  :op2 ( push :arg1 ( regulate
   :mod international :arg0-of ( stop
   :arg1 terrorist
   :arg2 ( use
    :arg1 ( information
    :arg2-of ( available :arg3-of free ))
   :arg2 ( and
    :op1 ( create :arg1 ( form
    :domain ( warfare
     :mod biology :example ( version
      :arg1-of modify :poss other_1 ) )
    :mod new ) )
    :op2 ( unleash :arg1 form  )
  ) ) ) ) ) ) ) ) )
```
REF: the report stated British government must help to stabilize weak states and push for international regulations that would stop **terrorists** using freely available information to create and unleash new forms of biological warfare such as a modified version of the influenza **virus** .

SYS: the report stated that the **Britain government** must help stabilize **the weak** states and push international regulations to stop the use of freely available information to create a form of **new** biological warfare such as **the modified** version of the influenza .

COMMENT: **coverage** , **disfluency**, **attachment**

```
state
 :arg0 ( person
 :arg0-of ( have-org-role
  :arg1 ( committee :mod technical )
  :arg3 ( expert
  :arg1 person
  :arg2 missile
  :mod loc_0 ) ) )
 :arg1 ( evidence
 :arg0 equipment
 :arg1 ( plan :arg1 ( transfer :arg1 ( contrast
  :arg1 ( missile :mod ( just :polarity - ) )
  :arg2 ( capable
  :arg1 thing
  :arg2 ( make :arg1 missile ) ) ) ) )
 :mod ( impeach :polarity - :arg1 thing )
 :mod ( refute :polarity - :arg1 thing ) )
```
REF: a technical committee **of Indian missile** experts stated that the equipment was unimpeachable and irrefutable **evidence of a plan to transfer not just missiles but missile-making capability**.

SYS: **a technical committee expert on the technical committee** stated that the equipment is **not impeach** , **but it is not refutes** .

COMMENT: **coverage** , **disfluency**, **attachment**

Figure 3: Linearized AMR after preprocessing, reference sentence, and output of the realizer. We mark, with colors, common error types: disfluency, coverage (missing information from the input graph), and attachment (implying a semantic relation from the AMR between incorrect entities).

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
