# Peer review of "Neural AMR: Sequence-to-Sequence Models for Parsing and Generation"

_ACL 2017 — decision unknown_

[Official Review · Reviewer 1 · rating 4 · confidence 4]
soundness 3 · originality 4 · clarity 3 · impact 4 · substance 4 · appropriateness 5 · meaningful comparison 4 · presentation format Poster

- Strengths:

The paper demonstrates that seq2seq models can be comparatively effectively
applied to the tasks of AMR parsing and AMR realization by linearization of an
engineered pre-processed version of the AMR graph and associated sentence,
combined with 'Paired Training' (iterative back-translation of monolingual data
combined with fine-tuning). While parsing performance is worse than other
reported papers (e.g., Pust et al., 2015), those papers used additional
semantic information. 

On the task of AMR realization, the paper demonstrates that utilizing
additional monolingual data (via back-translation) is effective relative to a
seq2seq model that does not use such information. (See note below about
comparing realization results to previous non-seq2seq work for the realization
task.)

- Weaknesses:

 At a high-level, the main weakness is that the paper aims for empirical
comparisons, but in comparing to other work, multiple aspects/dimensions are
changing at the same time (in some cases, not comparable due to access to
different information), complicating comparisons. 

For example, with the realization results (Table 2), PBMT (Pourdamghani et al.,
2016) is apparently trained on LDC2014T12, which consists of 13,051 sentences,
compared to the model of the paper, which is trained on LDC2015E86, which
consists of 19,572 sentences, according to http://amr.isi.edu/download.html.
This is used in making the claim of over 5 points improvement over the
state-of-the-art (PBMT) in line 28/29, 120/121, and line 595, and is only
qualified in the caption of Table 2. To make a valid comparison, the approach
of the paper or PBMT needs to be re-evaluated after using the same training
data.

- General Discussion:

Is there any overlap between the sentences in your Gigaword sample and the test
sentences of LDC2015E86? Apparently LDC2015E86 contains data from the ''proxy
report data in LDC's DEFT Narrative Text Source Data R1 corpus (LDC2013E19)''
(Accessible with LDC account: https://catalog.ldc.upenn.edu/LDC2015E86). It
seems LDC2013E19 contains data from Gigaword
(https://catalog.ldc.upenn.edu/LDC2013E19). Apparently AMR corpus LDC2014T12
also contained ''data from newswire articles selected from the English Gigaword
Corpus, Fifth Edition'' (publicly accessible link:
https://catalog.ldc.upenn.edu/docs/LDC2014T12/README.txt). Please check that
there is no test set contamination.

Line 244-249: Did these two modifications to the encoder make a significant
difference in effectiveness? What was the motivation behind these changes?

Please make it clear (in an appendix is fine) for replication purposes whether
the implementation is based on an existing seq2seq framework.

Line 321: What was the final sequence length used? (Consider adding such
details in an appendix.)

Please label the columns of Table 1 (presumably dev and test). Also, there is a
mismatch between Table 1 and the text: ''Table 1 summarizes our development
results for different rounds of self-training.'' It appears that only the
results of the second round of self-training are shown.

Again, the columns for Table 1 are not labeled, but should the results for
column 1 for CAMR instead be 71.2, 63.9, 67.3--the last line of Table 2 in
http://www.aclweb.org/anthology/S16-1181 which is the configuration for
+VERB+RNE+SRL+WIKI? It looks like the second from last row of Table 2 in CAMR
(Wang et al., 2016) is currently being used. On this note, how does your
approach handle the wikification information introduced in LDC2015E86? 

7.1.Stochastic is missing a reference to the example.

Line 713-715: This seems like a hypothesis to be tested empirically rather than
a forgone conclusion, as implied here.

Given an extra page, please add a concluding section.

How are you performing decoding? Are you using beam search?

As a follow-up to line 161-163, it doesn't appear that the actual vocabulary
size used in the experiments is mentioned. After preprocessing, are there any
remaining unseen tokens in dev/test? In other words, is the unknown word
replacement mechanism (using the attention weights), as described in Section
3.2, ever used? 

For the realization case study, it would be of interest to see performance on
phenomena that are known limitations of AMR, such as quantification and tense
(https://github.com/amrisi/amr-guidelines/blob/master/amr.md).

The paper would benefit from a brief discussion (perhaps a couple sentences)
motivating the use of AMR as opposed to other semantic formalisms, as well as
why the human-annotated AMR information/signal might be useful as opposed to
learning a model (e.g., seq2seq itself) directly for a task (e.g., machine
translation).

For future work (not taken directly into account in the scores given here for
the review, since the applicable paper is not yet formally published in the
EACL proceedings): For parsing, what accounts for the difference from previous
seq2seq approaches? Namely, between Peng and Xue, 2017 and AMR-only (as in
Table 1) is the difference in effectiveness being driven by the architecture,
the preprocessing, linearization, data, or some combination thereof? Consider
isolating this difference. (Incidentally, the citation for Peng and Xue, 2017
[''Addressing the Data Sparsity Issue in Neural AMR Parsing''] should
apparently be Peng et al. 2017
(http://eacl2017.org/index.php/program/accepted-papers;
https://arxiv.org/pdf/1702.05053.pdf). The authors are flipped in the
References section.

Proofreading (not necessarily in the order of occurrence; note that these are
provided for reference and did not influence my scoring of the paper):

outperform state of the art->outperform the state of the art

Zhou et al. (2016), extend->Zhou et al. (2016) extend

(2016),Puzikov et al.->(2016), Puzikov et al.

POS-based features, that->POS-based features that

language pairs, by creating->language pairs by creating

using a back-translation MT system and mix it with the human
translations.->using a back-translation MT system, and mix it with the human
translations.

ProbBank-style (Palmer et al., 2005)->PropBank-style (Palmer et al., 2005)

independent parameters ,->independent parameters,

for the 9.6% of tokens->for 9.6% of tokens

maintaining same embedding sizes->maintaining the same embedding sizes

Table 4.Similar->Table 4. Similar

realizer.The->realizer. The

Notation: Line 215, 216: The sets C and W are defined, but never subsequently
referenced. (However, W could/should be used in place of ''NL'' in line 346 if
they are referring to the same vocabulary.)

[Official Review · Reviewer 2 · rating 4 · confidence 4]
soundness 3 · originality 4 · clarity 3 · impact 4 · substance 3 · appropriateness 5 · meaningful comparison 4 · presentation format Oral Presentation

The authors use self-training to train a seq2seq-based AMR parser using a small
annotated corpus and large amounts of unlabeled data. They then train a
similar,
seq2seq-based AMR-to-text generator using the annotated corpus and automatic
AMRs produced by their parser from the unlabeled data. They use careful
delexicalization for named entities in both tasks to avoid data sparsity. This
is the first sucessful application of seq2seq models to AMR parsing and
generation, and for generation, it most probably improves upon state-of-the
art.

In general, I really liked the approach as well as the experiments and the
final performance analysis.
The methods used are not revolutionary, but they are cleverly combined to
achieve practial results.
The description of the approach is quite detailed, and I believe that it is
possible to reproduce the experiments without significant problems.
The approach still requires some handcrafting, but I believe that this can be
overcome in the future and that the authors are taking a good direction.

(RESOLVED BY AUTHORS' RESPONSE) However, I have been made aware by another
reviewer of a data overlap in the
Gigaword and the Semeval 2016 dataset. This is potentially a very serious
problem -- if there is a significant overlap in the test set, this would
invalidate the results for generation (which are the main achievemnt of the
paper). Unless the authors made sure that no test set sentences made their way
to training through Gigaword, I cannot accept their results.

(RESOLVED BY AUTHORS' RESPONSE)  Another question raised by another reviewer,
which I fully agree with, is the 
5.4 point claim when comparing to a system tested on an earlier version of the
AMR dataset. The paper could probably still claim improvement over state-of-the
art, but I am not sure I can accept the 5.4 points claim in a direct comparison
to Pourdamghani et al. -- why haven't the authors also tested their system on
the older dataset version (or obtained Pourdamghani et al.'s scores for the
newer version)?

Otherwise I just have two minor comments to experiments: 

- Statistical significance tests would be advisable (even if the performance
difference is very big for generation).

- The linearization order experiment should be repeated with several times with
different random seeds to overcome the bias of the particular random order
chosen.

The form of the paper definitely could be improved.
The paper is very dense at some points and proofreading by an independent
person (preferably an English native speaker) would be advisable. 
The model (especially the improvements over Luong et al., 2015) could be
explained in more detail; consider adding a figure. The experiment description
is missing the vocabulary size used.
Most importantly, I missed a formal conclusion very much -- the paper ends
abruptly after qualitative results are described, and it doesn't give a final
overview of the work or future work notes.

Minor factual notes:

- Make it clear that you use the JAMR aligner, not the whole parser (at
361-364). Also, do you not use the recorded mappings also when testing the
parser (366-367)?

- Your non-Gigaword model only improves on other seq2seq models by 3.5 F1
points, not 5.4 (at 578).

- "voters" in Figure 1 should be "person :ARG0-of vote-01" in AMR.

Minor writing notes:

- Try rewording and simplifying text near 131-133, 188-190, 280-289, 382-385,
650-659, 683, 694-695.

- Inter-sentitial punctuation is sometimes confusing and does not correspond to
my experience with English syntax. There are lots of excessive as well as
missing commas.

- There are a few typos (e.g., 375, 615), some footnotes are missing full
stops.

- The linearization description is redundant at 429-433 and could just refer to
Sect. 3.3.

- When refering to the algorithm or figures (e.g., near 529, 538, 621-623),
enclose the references in brackets rather than commas.

- I think it would be nice to provide a reference for AMR itself and for the
multi-BLEU script.

- Also mention that you remove AMR variables in Footnote 3.

- Consider renaming Sect. 7 to "Linearization Evaluation".

- The order in Tables 1 and 2 seems a bit confusing to me, especially when your
systems are not explicitly marked (I would expect your systems at the bottom).
Also, Table 1 apparently lists development set scores even though its
description says otherwise.

- The labels in Table 3 are a bit confusing (when you read the table before
reading the text).

- In Figure 2, it's not entirely visible that you distinguish month names from
month numbers, as you state at 376.

- Bibliography lacks proper capitalization in paper titles, abbreviations and
proper names should be capitalized (use curly braces to prevent BibTeX from
lowercasing everything).

- The "Peng and Xue, 2017" citation is listed improperly, there are actually
four authors.

***
Summary:

The paper presents first competitive results for neural AMR parsing and
probably new state-of-the-art for AMR generation, using seq2seq models with
clever
preprocessing and exploiting large a unlabelled corpus. Even though revisions
to the text are advisable, I liked the paper and would like to see it at the
conference. 

(RESOLVED BY AUTHORS' RESPONSE) However, I am not sure if the comparison with
previous
state-of-the-art on generation is entirely sound, and most importantly, whether
the good results are not actually caused by data overlap of Gigaword
(additional training set) with the test set.

***
Comments after the authors' response:

I thank the authors for addressing both of the major problems I had with the
paper. I am happy with their explanation, and I raised my scores assuming that
the authors will reflect our discussion in the final paper.